# Microbial Characterization of Retail Cocoa Powders and Chocolate Bars of Five Brands Sold in Italian Supermarkets

**DOI:** 10.3390/foods11182753

**Published:** 2022-09-07

**Authors:** Lucilla Iacumin, Michela Pellegrini, Andrea Colautti, Elisabetta Orecchia, Giuseppe Comi

**Affiliations:** Department of Agricultural, Food, Environmental and Animal Sciences, University of Udine, Via Sondrio 2/a, 33100 Udine, Italy

**Keywords:** cocoa powder, chocolate, microorganisms, spoilage, growth

## Abstract

A microbial characterization of cocoa powder and chocolate bars of three batches of five different brands sold in Italian markets was performed. The results showed a variable microbial population consisting of mesophilic and thermophilic spore formation in both types of products. The chocolate bars were also contaminated with molds of environmental origin. *Bacillus* spp. and *Geobacillus* spp. were found in both products. The chocolate bars were also contaminated by molds belonging to the genera *Penicillium* and *Cladosporium*. The sporogenous strains mainly originate from the raw materials, i.e., cocoa beans, as the heat treatments involved (roasting of the beans and conching of the chocolate) are not sufficient to reach commercial sterility. Furthermore, the identified spore-forming species have often been isolated from cocoa beans. The molds isolated from chocolate seem to have an origin strictly linked to the final phases of production (environment and packaging). However, the level of contaminants is limited (<2 log CFU/g); the molds do not develop in both products due to their low Aw (<0.6) and do not affect the safety of the products. However, a case of mold development in chocolate bars was observed. Among the isolated molds, only *Penicillium lanosocoeruleum* demonstrated a high xero-tolerance and grew under some conditions on chocolate bars. Its growth could be explained by a cocoa butter bloom accompanied by the presence of humidity originating from the bloom or acquired during packaging.

## 1. Introduction

Chocolate is derived from the seeds of the cocoa tree whose origin seems to date back to pre-Columbian civilizations. The Mayans and Aztecs used cocoa seeds as bargaining chips or money and burned them together with incense during religious rituals. For the Mayans, chocolate was a cocoa drink prepared with hot water flavored with chili and pepper, known as “the food of the gods”, which was intended only for certain social classes, such as sovereigns, nobles and warriors [1]. In Europe, it was introduced by the Spanish, who modified its composition, replacing chili with vanilla and sugar to overcome the intrinsic bitter taste. Currently, the production chain of chocolate is very complex and begins with the natural fermentation of cocoa beans (*Theobroma cacao*) [2], followed by drying, roasting, removal of skins, grinding and alkalinization to improve the organoleptic characteristics. This process leads to the production of cocoa mass (liquor), which is squeezed to separate the cocoa butter from the powder. The latter is partially defatted at levels between 10% and 20% (dry matter) [3,4]. The powder obtained constitutes cocoa, while the cocoa butter is mixed with different ingredients [5] to form a homogeneous dispersion of solid cocoa and crystals of sugar in a continuous phase, consisting of fat crystals and liquid fat [6] and to obtain the crystallization of cocoa butter in the correct polymorphic V form. Then, the cocoa butter is added to other ingredients (e.g., sugar, cocoa mass, milk powder, or hazelnut paste) and the mixture is refined with conching, obtained at 50–80 °C, to reduce acidity; all ingredients are amalgamated and the particle size is decreased to allow chocolate to melt in the mouth and release aromas [1]. Subsequently, the chocolate is cooled from 45 °C to approximately 30 °C depending on the type of chocolate (milk or dark) in a phase called tempering to restore the correct polymorphic form [6]. Then, the liquid chocolate is placed in polycarbonate molds and cooled to allow the solidification of the fat phase in the chocolate mass to obtain the correct crystallization, which leads to the easier removal of the solidified chocolate from the mold. The chocolate is finally packed; due to its tendency to absorb odors and moisture, it must be wrapped in tinfoil or closed in an airtight box in equilibrium relative humidity (ERH) at approximately 35–40% [6]. Over the centuries, the production chain of chocolate has evolved considerably from the introduction of dark chocolate bars in 1847 by Joseph Fry, the first solid milk chocolate in 1876 by Daniel Peter and the production of softer chocolate with a better taste in 1880 by Rodolphe Lindt [5]. Subsequently, the chocolate production process has undergone several changes to achieve better sensory quality and increased productivity and meet the growing demands for chocolate products. Throughout the production process, various microorganisms can contaminate the raw materials (cocoa beans), intermediates (cocoa powder or butter) and finished product (chocolate). Cocoa beans are contaminated by a high number of microorganisms and the level of contamination may increase during postharvest and industrial processing operations. In particular, the latter are unable to eliminate contaminants as these mainly consist of *Bacillus* spp. [7], which are known to produce heat-resistant spores. However, they rarely lead to spoilage and safety issues [8] considering the levels of water activity (Aw) of cocoa powder and chocolate. Studies investigating the microbial ecology of commercial cocoa powder revealed the presence of an aerobic microbial population ranging from 2.0 to 5.0 log CFU/g, which included *Bacillus licheniformis, B. cereus*, *B. megaterium* and *B. subtilis* [9]. In other studies, spores of *B. subtilis* and *B. licheniformis* were isolated in variable percentages between 36% and 83% of the total isolates [10,11]. Filamentous fungi, including xerophilous strains belonging to the species *Aspergillus glaucus* [12] and *Crisosporium* spp. [13], can also contaminate cocoa beans. Fungal contamination in the cocoa powder is very limited due to the industrial process operations applied, which solely allows the survival of bacterial spores [7]. Conversely, in chocolate and chocolate-based products, microbial contamination may occur more frequently during and after their production [14], especially during the packaging phase. Such contamination has been demonstrated by different foodborne outbreaks [15] caused by *Salmonella enterica* subsp. *enterica* serotypes. Consequently, it is suggested that filamentous fungi can also become common components of the microbial population in the production process and, thus, survive heat and drying [13]. In particular, *Bettsia alvei* (teleomorph: *Chrysosporium farinicola*) and *Chrysosporium xerophilum* were isolated from chocolate with hazelnuts and *Neosortorya glabra* was isolated from spoiled chocolate [16]. Molds belonging to the species *C. xerophilum*, *C. inops* and *C. farinicola* have been isolated from commercial chocolate bars with an Aw of approximately 0.28 [13]. Fungi, as opposed to bacteria, are tolerant of low Aw and pH, ethanol and other preservatives and, therefore, are held accountable for possible alterations [13]. Spoilage has been observed in classic chocolate bars and products in which chocolate is used as an edible film or filling, such as pralines, which are made of solid chocolate stuffed with a soft filling (sometimes liquid), consisting of butter, liquor, sugar, fruit, nuts or marzipan [17]. Praline fillings contain a high amount of sugar or other dissolved solutes, which reduce the Aw below the levels needed to support the growth of most microorganisms. The microbial stabilization of chocolate filling is often carried out using preservatives, such as sorbic and benzoic acid and alcohol. However, it is not always enough to prevent the growth of osmophilic yeasts, xerophilous molds and bacteria, which tolerate low Aw and may cause several damages in the confectionery industry, such as off-flavor, slime and gas production, leading to the breaking of pralines or the presence of a liquid fraction on the surface [17,18]. Spoilage in classic chocolate bars consists of unpleasant tastes and smells and the formation of superficial slime. Several authors have described numerous alterations in pralines caused by molds [18] and osmophilic yeasts belonging to the genus *Zygosaccharomyces* [17]; the latter is often associated with the deterioration of products with a high sugar content [19,20,21]. Classic chocolate bars have an Aw less than 0.5; therefore, yeasts and molds that survived the production process or resulted from contamination cannot develop under good postproduction storage conditions. It has been observed that the critical phase in chocolate production that prevents the development of molds is represented by the cooling and solidification phases. It is necessary to prevent the formation of moisture on the chocolate surface to avoid sugar blooms and the development of molds. Beckett [5] recommends keeping the cooling temperature precisely above the dew point to prevent the formation of condensation. In fact, by avoiding the use of a too low cooling temperature, cocoa butter is prevented from settling in an incorrect crystalline form, which brings a loss of moisture and prevents the de-molding of the chocolate. The humidity of the manufacturing environment can influence the chemical, physical and microbiological stability of chocolate. Excessive humidity can negatively influence the viscosity of chocolate and affect microbial growth. To eliminate moisture on the surface or prevent chocolate from absorbing moisture, it is recommended to keep the relative moisture balance (ERH) at approximately 35–40% during the production process. The present work aimed to study the microbial population of cocoa powder and chocolate bars produced by five different European companies, verifying the ability of the isolates to develop into model systems and identifying the fungal species involved in spoilage.

## 2. Materials and Methods

### 2.1. Sample Collection and Microbiological Analysis

Three different lots of 5 brands of cocoa powder and chocolate bars were analyzed. The samples (50 per lot), which are produced by the most representative European companies, were purchased in their original packages at different markets in the region of Lombardy in northern Italy. Each lot of chocolate bars included 20 samples of extra bitter chocolate (EBC; Aw 0.510 ± 0.006) made with sugar, cocoa mass, cocoa butter, emulsifier (soya lecithin) and natural flavor (cocoa: 52% minimum); 20 samples of extra fine milk chocolate (EFMC; Aw 0.530 ± 0.002) made with sugar, whole milk powder, cocoa butter, cocoa mass and emulsifier (soya lecithin) (cocoa: 30% minimum); 10 samples of milk chocolate filled with milk cream with added vanilla flavor (MCFC; Aw 0.550 ± 0.003) made with sugar, vegetable fats, milk powder, cocoa butter, wheat flour, cacao mass and emulsifier (soya lecithin) and filled with glucose and fructose syrup, nut mass, raising agent (E501, E503 and E500), salt, natural flavor, acidifier (E524) and skimmed milk powder. Five additional moldy MCFC bars collected in a small shop were also analysed.

The samples were diluted in sterile saline-peptone water (10 g/L glucose, 8 g/L NaCl, 1 g/L bacteriological peptone, Oxoid, Milan, Italy) at a ratio of 1:5, but each dilution was inoculated in 5 plates of the different culture media to reduce the lower detection limit at 1 CFU/g. The addition of sugar eliminates the risk of the inactivation of any osmophilic or osmotolerant species present in the samples. One milliliter of each dilution was inoculated into different culture media to quantify and isolate the microorganisms present. Yeasts and molds were detected in malt extract agar (MEAG, Oxoid, Milan, Italy, supplemented with 100 g/L glucose and Chloramphenicol 0.100 g/L, Sigma–Aldrich, Milan, Italy) incubated at 25 °C for 3–5 days. The bacteria were tested in Plate Count Agar (PCAG, Oxoid, Milan, Italy, supplemented with 100 g/L glucose) incubated at 30 °C (mesophiles) and 55 °C (thermophiles) for 48–72 h, *Enterobacteriaceae* was tested in Violet Red Bile Glucose Agar (VRBGA, Oxoid, Milan, Italy) incubated at 37 °C for 48 h and fecal coliforms were tested in Violet Red Bile Lactose Agar (VRBLA, Oxoid, Milan, Italy) incubated at 43.5 °C for 48 h. The *Enterobacteriaceae* and fecal coliforms were presumptively identified by forming purple-pink-colored colonies on VRBGA and VRBLA, respectively. Additionally, the total aerobic spore number was determined. Briefly, the first dilution was pasteurized at 80 °C for 10 min in a water bath and then, after chilling, 0.1 mL of this and subsequent dilution were plated in PCAG. For *Listeria* spp. ISO 11290-1:1996 Adm.1:2004 (Microbiology of food and animal feeding stuff–Horizontal method for the detection of *Listeria monocytogenes*) and *Salmonella* spp. ISO 6579-1: 2002 Cor.1:2004 (Microbiology of food and animal feeding stuff–Horizontal method for the detection of *Salmonella* spp.) were performed. *Listeria monocytogenes*—Briefly–25 g product were added to 225 mL of Fraser broth (Oxoid, Italy) incubated at 30 °C for 24 h, then an aliquot of this broths was streaked on Chromocult Listeria Agar according to Ottaviani/Agosti agar (Biolife, Milan, Italy) and incubated at 37 °C for 24 h. On this, agar *L. monocytogenes* produce typical blue-green colonies surrounded by an opaque halo); *Salmonella* (briefly: 25 g product were added to 225 mL of Buffered Peptone Water (BPW, Oxoid, Italy) was incubated 18 h at 37 °C, then 1 mL of BPW in 9 mL of Rappaport Vassiliadis broth (RVB, Oxoid, Milan, Italy) incubated at 42 °C for 18–24 h. An aliquot of RVB was streaked on Xylose Lysine Tergitol 4 agar (Oxoid, Milan, Italy) incubated at 37 °C for 24 h. On this agar the black or center black colonies were presumptive *Salmonella*.

### 2.2. Microbial Identification

From the PCA plates of the cocoa powder and chocolate samples, 100 colonies, selected independently for their morphology, color and size, were isolated from each brand considered. After purification, the colonies were identified according to the PCR-DGGE (PCR-denaturing gradient gel electrophoresis) method as described by Iacumin et al. [22]. Briefly, amplicons to be subjected to DGGE analysis were obtained using primers P1 (5′-GCGGCGTGCCTAATACATGC-3′) and P4 (5′ ATCTACGCATTTCACCGCTAC-3′) spanning the V3 region of the 16S rDNA [23]. A GC-clamp (5′-CGC CCGCCGCGCCCCGCGCCCGTCCCGCCGCCCCCGCCCG-3) was attached to the 5′ end of primer P1. Amplifications were carried out in a final volume of 25 μL, containing 1 μL (100 ng total) template DNA, 10 mM Tris HCl (pH 8.3), 50 mM KCl, 1.5 mM MgCl2, 0.2 mM deoxynucleoside triphosphates (dNTPs), 1.25 U Taq polymerase (Invitrogen, Milan, Italy) and 0.2 μM each primer, using the C1000 Touch Thermal Cycler (Bio-Rad, Milan, Italy). The amplification cycle included an initial denaturation step at 95 °C for 5 min, followed by 35 series composed by denaturation, performed at 95 °C for 1 min, annealing at 45 °C for 1 min and extension performed at 72 °C for 1 min. Finally, an extension cycle, performed at 72 °C for 7 min, was added. Electrophoresis was performed in a 0.8-mm-thick polyacrylamide gel (8% (wt/vol) acrylamide bisacrylamide (37.5:1)), with a denaturing gradient from 30% to 50% (100% corresponded to 7 M urea and 40% (wt/vol) formamide) increasing in the direction of the electrophoretic run using the Dcode universal mutation detection system (Bio-Rad, Hercules, CA, USA). Gels were subjected to a constant voltage of 130 V for 3 h and 30 min at 60 °C. After the electrophoresis, gels were stained for 30 min in 1.25X Tris-acetate-EDTA containing 1X SYBR Green (final concentration; Molecular Probes, Milan, Italy). Pictures of the gels were visualized under UV light using the Syngene G: Box Chemi-XX9 (Syngene, Cambridge, UK) and digitally captured by using the software GeneSys version 1.5.7.0 (Syngene, Cambridge, UK). Strains with the same DGGE profile were grouped and 2 representatives per group were amplified using the primers P1 and P4 targeting 700 bp of the V1-V3 region of the 16S rRNA gene (rDNA). Amplifications were carried out as reported above. After purification, amplicons were sent to a commercial facility for sequencing (Eurofins AG, Ebersberg, Germany). The sequences were aligned in GenBank using the Blast program version 2.13.0 [24] to determine the closest known relatives of the partial 16S rDNA sequence obtained.

From the MEAG plates of each chocolate brand, 24 mold colonies were isolated and transplanted into the following three different medium cultures: Czapek Dox Agar (Oxoid, Milan, Italy), MEA and Malt Salt agar (5% malt extract; 5% NaCl; pH 6.2; Oxoid, Milan, Italy). The isolated molds were identified according to the traditional methods proposed by Samson et al. [25], by examination of macroscopic (colonial) and microscopic characteristics of the molds grown on the three agars. In particular, morphology, shape, size and color of the colonies and spores, production of pigment, as well as type, development and texture of the mycelium were observed. The presumptive identification was then confirmed by sequencing a fragment of ~600 bp of the D1-D2 region of the large-subunit rRNA gene according to the methods reported by Iacumin et al. [22]. Primers NL1 (5′-GCCATATCAATAAGCGGAAAAG-3′) and NL4 (5′-GGTCCGTGTTTCAAG ACGG-3′) were used. Reaction mixture and amplification protocol were the same as those described above. Sequence comparisons were performed in GenBank using the Blast program version 2.13.0.

### 2.3. Inocula Preparation

Bacteria: A single colony of each strain, purified in PCA, was inoculated in brain heart infusion broth (BHI, Oxoid, Milan, Italy) and incubated overnight at 30 °C (mesophiles) and 55 °C (thermophiles). Then 4 mL of the overnight culture of each strain (Table 1 were subjected to centrifugation at 13,400 rpm for 10 min at room temperature; next, the pellets were resuspended in saline-peptone water (Peptone, 1 g, Oxoid, Milan, Italy; NaCl, 30 g, Sigma–Aldrich, Milan, Italy; distilled H_2_O, 1000 mL; Aw 0.97). Each suspension was standardized at an optical density at 600 nm (OD_600_) of 0.1 and their microbial concentrations were evaluated in Brain Heart Infusion Agar (Oxoid, Milan, Italy), which was incubated at 30 °C (mesophiles) and 55 °C (thermophiles) for 48 h. The colonies were counted and each suspension contained approximately 10^7^ CFU/mL.

Mold: The mold spores, which were obtained by 5 days of culture (Table 2) in malt agar incubated at 25 °C, were collected and diluted in sterile peptone water (Peptone Oxoid 0.1%; NaCl 0.7%; Glucose 50%; distilled water 500 mL; pH 6.0, Aw 0.85). Briefly: 3 mL of peptone water was liberally spread on the colonies and the suspension was recovered with a Pasteur pipette. The presence of mycelia was checked by the hemocytometer. Only the suspension with a small part of the mycelium (<10 mycelium part/mL) was chosen. Each spore suspension was counted by an hemocytometer, reaching a concentration of approximately 10^7^ spores/mL.

Each mold or bacterial suspension was obtained by mixing three different strains of each species.

### 2.4. Growth in Agar under Stressful Conditions

Growth in substrates with different stress factors was evaluated according to the modified Vermeulen et al. [21] method. Briefly, MEA and PCA were added to 50% glucose/fructose (w/w) at a 1:1 ratio and the pH was adjusted to different pH levels (2.0, 3.0, 4.0, 5.0, 6.0 and 7.0) using lactic acid. Glycerol was then added until the following values of Aw were reached: 0.65, 0.70, 0.75, 0.80, 0.85 and 0.90. Then, 1 μL of each diluted suspension (7 log CFU/mL) obtained as described in 2.3 was inoculated in the center of each agar, which was incubated at 25 °C (mold strains) and 30–55 °C (sporogenous strains).

Growth was measured after 50 days; for molds, the diameter of the colonies grown in MEA was measured, while for the sporogenous colonies the presence of the colony in PCA was evaluated in comparison with the same inoculated plates incubated at 0 °C.

### 2.5. Growth of Mold Intentionally Inoculated in a Chocolate Model System Packaged with Aluminized Paper [26]

Samples of each typology of chocolates as described in Section 2.1 were dissolved in a water bath to obtain a paste, which was treated with different rates of distilled water to increase the Aw values as shown in Table 3. Then, the paste was placed in molds to obtain 80 bars for each type of chocolate. Each bar was inoculated with 100 μL of a suspension of *P. lanosocoeruleum* at a final concentration of approximately 100 spores/cm^2,^ as reported in Section 2.3. After the inoculation, the bars of each Aw were divided into the following two groups: the first group (10 bars) was packaged under vacuum (VP) in an aluminized paper and the second group (10 bars) was packaged in aluminized paper with air remaining between the bars and the paper (Table 3). Then 5 bars of each packaging (under vacuum and air) were incubated at 20 °C and 5 at 30 °C. Every 10 days until the end of their shelf life (10 months), the packaged samples were opened and analyzed to evaluate mold growth and then repackaged and restored at the tested temperatures. Positive results were indicated on the day we observed the presence of mold growth (visible hyphae or spores) on the chocolate surface bars. The grown molds were isolated in malt agar and identified as indicated in Section 2.2. The spore-forming bacteria were not tested because they could not grow at less than 0.90 Aw.

### 2.6. Growth of Mold Intentionally Inoculated in Chocolate Stored in Different Equilibria Relative Humidity (ERH) at 4, 20 and 30 °C

Six solutions were used, corresponding to ERH 99, 94, 90, 86, 81 and 75%. The NaCl concentration and the corresponding ERH are listed in Table 4. In particular, the first solution was represented by H_2_O (ERH% 99), while the last solution was a supersaturated solution of sodium chloride (ERH% 75). Then, the solutions were posed in boxes, including a grid/support on which the chocolate bars were leaned to avoid direct contact between the solutions and the chocolate bar samples. Three different chocolate bars (extra bitter chocolate, Aw 0.510 ± 0.006; extra fine milk chocolate, 0.530 ± 0.02; and milk chocolate filled with milk cream with added vanilla flavor, 0.550 ± 0.003) weighing 30 ± 2 g were inoculated by swabs with 100 μL of a diluted suspension of *P. lanosocoeruleum* obtained as described in Section 2.3 at a final concentration of approximately 100 spores/cm^2^ and then placed on the grids inside hermetic boxes in the presence of different solutions of NaCl. The boxes were closed and stored at 4, 20 (room temperature) and 30 °C in the dark to eliminate the risk of fat photooxidation. In total, 162 boxes were used. Each chocolate bar was closed in 6 different boxes containing the ERH solution and kept at three temperatures. The analysis was made in triplicate. Each day until the end of their shelf life (10 months), the boxes were inspected to evaluate mold growth. Positive results were indicated on the day we observed the presence of mold growth (visible hyphae or spores) on the chocolate surface bars. The growth molds were isolated in malt agar and identified as indicated in Section 2.2.

The spore-forming bacteria were not tested because they could not grow at less than 0.90 Aw.

### 2.7. Physico-Chemical Analysis

The pH was determined using a pH meter (Basic 20, Crison Instruments, Alella, Spain) and the water activity (Aw) and equilibrium relative humidity (ERH) determination were performed using an Aqualab 4 TE (Decagon Devices, Inc., Pullman, WA, USA).

### 2.8. Statistical Analysis

The data were analyzed using Statistica 7.0 vers. 8 software (Statsoft, Inc., Tulsa, OK, USA, 2008). The values of the different parameters were compared by one-way analysis of variance and the means were then compared using Tukey’s honest significance test. The differences were considered significant at *p* < 0.05.

## 3. Results and Discussion

Microbial, chemical and physical analyses of samples of three batches of five different brands of cocoa powder and chocolate bars sold in the Italian market were performed. The reported results show the means and the standard deviations of all samples of each brand analyzed (Table 5, Table 6 and Table 7). The chemico-physical analysis showed that the products have low Aw levels and are able to avoid any microbial growth, including xerophilic and xero-tolerant microorganisms. The statistical analysis revealed a significant difference in the Aw among the brands; however, this difference does not affect the stability of the different samples. In fact, the Aw in cocoa powders, regardless of the batches and brands analyzed, was always lower than 0.42, while, regarding the chocolate analyzed, the Aw was always lower than 0.55 (Table 5).

The pH values varied more widely than the Aw values; these values ranged from 5.3 to 7.1 in cocoa powder and from 5.5 to 7.0 in chocolate (Table 5). Even in this case, the differences did not influence the stability of the products, which was determined by the Aw level. Most likely, the differences observed among the brands reflect the choices of raw materials, including the cocoa varieties of beans and the production techniques. The varieties of cocoa present in the *Theobroma* genus include more than 20 spontaneous species cultivated and used in restricted areas by indigenous populations to produce tonic and corroborating derivatives and cocoa substitutes. Currently, the following three large botanical groups (varieties) of cocoa are recognized: Criollo (or finos) from *Theobroma cacao*, Forastero (or amazonicos) from *Theobroma cacao sphaerocarpum and* Trinitario, a hybrid of the first two. Each cocoa variety is used in blends and rarely in a single variety (cru). For example, Criollo cocoa is often used as a reinforcer of other blends with a weak and nonpersistent aroma; Forastero cocoa (i.e., standard cocoa) is mainly used by large retailers; and Trinitario cocoa has a fruity and persistent aroma and is always used in a blend [1]. However, even though each chocolate producer uses blends of particular varieties to endow chocolate with flavor, persistence of aroma and body, notably, for 40 years, chocolates prepared with varieties used in pure form have appeared on the market and are currently of particular commercial interest [1].

### 3.1. Cocoa Powder Contamination

The cocoa powders evaluated do not have considerably different microbiological qualities. Table 6 shows that the different samples analyzed had contamination levels lower than 2 log CFU/g. In contrast, in samples 4 and 5 (Table 6), thermophilic microorganisms capable of developing at 55 °C were below the detection limit of 1 CFU/g. Conversely, in the first three cocoa brands, thermophiles and spore-forming mesophilic bacteria were found, but no significant difference (*p* > 0.05) was noted between the two spore-forming groups. Finally, the concentration of the total bacterial count (CBT), which includes both spore-forming and non-spore-forming bacteria, was similar and not significantly different from the concentration (*p* > 0.05) of the spore-forming bacteria. Therefore, this population must be included in the spore-forming total count. The data concerning the microbial concentrations confirm the reports by Lima et al. [10], who conducted a study to evaluate the microbiota of cocoa powder with particular regard to the presence of heat-resistant and thermophilic spore-forming bacteria. Conversely, the CBT was less than 2 and 4 log CFU/g as reported by Gabis et al. [9].

As shown in Table 6, yeasts and the main pathogens, such as *Salmonella* spp., *Listeria monocytogenes,* fecal coliforms and Enterobacteriaceae were not detected. In fact, they are eliminated during the production process and are not spore-forming bacteria as demonstrated by various authors [4,27].

The spore-forming species isolated from cocoa include *Bacillus licheniformis*, *B. subtilis*, *B. amyloliquefaciens* and *Geobacillus stearothermophilus* (Table 8). There was no significant difference in the species among the brands considered, except for *B. licheniformis*, which was isolated in three brands (2, 4 and 5). A clear predominance of *B. subtilis* was observed (58.2%); This species is heat resistant and, therefore, can survive the temperatures used in the production process, particularly during roasting; *B. subtilis* was followed by *Geobacillus stearothermophilus* (19.2%), which is the only true thermophile, *B. amyloliquefaciens* (17.4%) and finally *B. licheniformis* (5.2%). The predominance of *B. subtilis* and the presence of *B. licheniformis* have already been highlighted in other works [17,28]. In particular, in previous works, both strains along with *Bacillus cereus* and *Bacillus megaterium* represented the main population, surviving heat treatments in cocoa powder production [9,11].

Indeed, both *B. subtilis* and *B. licheniformis* are known for the wide variability of the properties of their spores and especially for their high resistance to heat treatments [10,29,30]. Some strains have shown decimal reductions similar to those of *G. stearothermphilus*, which is considered a typical spoilage microorganism of canned products [28,31]. The quantification of heat-resistant spores in cocoa powder is extremely important, especially when it is used to produce cocoa beverages [32]. In fact, since they are stored at room temperature, such beverages could undergo spoilage due to the activity of these spores, which survive the adopted heat treatment processes [32]. Cocoa powder is believed to be a good reservoir of bacilli and geobacilli [33]. In particular, *G. stearothermophilus* and other bacilli isolated in this study are often isolated from fermenting cocoa beans [2,34], although their function has not yet been elucidated. According to Lima et al. [10], *G. steratothermophilus* represents sporadic contamination because the participation of thermophiles in bean fermentation should be considered rare, although bean fermentation lasts approximately 6–7 days after fruit opening and reaches temperatures that can exceed 50 °C [2,35]. Indeed, the activities of thermophilic bacteria, such as *G. steratothermophilus,* have recently been studied. It has been demonstrated that they can grow when the temperature of cocoa bean fermentation reaches 50 °C or higher, but they do not influence the consuming sugars, producing ethanol, organic acids and volatile compounds of indigenous microorganisms [33,36].

The bacterial strains isolated from cocoa powders all belong to spore-forming species and are completely different from those that predominate during bean fermentation [2,37,38]. In fact, the latter are represented by lactic acid bacteria (LAB), yeasts and molds. The development of these microbial groups occurs in sequence and strictly depends on the chemical-physical characteristics of the beans. Their sequential development is crucial for the development of the particular characteristics highlighted in the finished product [2]. Fermentation involves the following two phases: anaerobic and aerobic [39]. During the anaerobic phase, which lasts 48–72 h, yeasts and LAB predominate. The yeasts *Saccharomyces cerevisiae*, *Pichia kudriavzevii*, *P. manshurica* and *Hanseniaspora opuntiae* [40] ferment the sugars present (glucose and fructose) and some organic acids, such as citric acid and malic acid, releasing ethanol and increasing the temperature [41]. Among LAB, *Lactiplantibacillus plantarum* and *Limosilactobacillus fermentum* are found, which coexist with species belonging to the genera *Leuconostoc* and *Lactococcus*. These microbial groups ferment organic acids into lactic and acetic acid and mannitol, lowering the pH and promoting microbial stability and color by releasing aromatic precursors [41,42]. Furthermore, several microorganisms, such as *Pichia kudriavzevii* and *Bacillus* spp., degrade the pulp through the production of pectolytic enzymes [43,44], which lose moisture and acquire oxygen, favoring the aerobic phase. During this second phase, *Acetobacter* oxidizes ethanol into acetic acid, inactivates the germination of beans and increases the permeability of the membrane, allowing the release of the precursor of aromatic compounds [45,46,47]. During fermentation, a temperature increase occurs, which leads to the degradation of phenolic compounds, such as catechins and epicatechins and methylxanthines (caffeine, theobromine and theophylline) and the inactivation of oxidase [48]. The degradation of these compounds reduces the bitter taste and astringency of the beans [46] and increases the vasodilatory, diuretic and relaxing effects of cocoa [49]. Parallel to the fermentation, drying and storage of beans, various fungal groups belonging to beneficial species, such as *Absidia corymbifera*, *A. flavus* and *Penicillium paneum and* yeasts and potentially harmful *A. flavus*, *A. parasiticus, A. nomeus*, *A. niger* group, *A. carbonarius* and *A. ochraceus* groups responsible for the production of mycotoxins. In particular, more than 20 fungal genera have been reported and divided into the following two groups: sensitive or resistant to the presence of acetic acid, which is due to the increase in temperature caused by temperature/oxidation evaporation and does not completely carry out its antimicrobial function. However, acid-tolerant molds do not develop due to dehydration, fermentation and storage, which tend to reduce or eliminate fungal contamination [50], but recontamination is possible. Dehydration introduces a further problem linked to the development of xerophilic molds, which can cause off-odor and off-flavor production. Therefore, there is a tendency to accelerate the dehydration phase and remove air during the storage of fermented beans. Despite the presence of a multi-microbial population in fermented beans, all cocoa samples analyzed here show solely the presence of spore-forming microorganisms, probably due to the treatments the beans undergo post-fermentation. A crucial phase is bean roasting, which is performed to extract the compounds that define the aromatic components of cocoa and chocolate. This heat treatment is carried out in batch or continuous systems at temperatures ranging between 110 and 150 °C for 15 min to 10 h or upon reaching a humidity level of approximately 3% [9,51,52]. It can be concluded that roasting eliminates all non-spore-forming microorganisms, such as molds, yeasts and bacteria and reduces the spore-forming component belonging to the *Bacillus* genus [4,10,32,36]. It is known that the heat resistance of the spores of *Bacillus* and *Geobacillus* is variable and strictly depends on the species, strain and matrix characteristics, such as pH and Aw. Several authors have demonstrated that roasting, even if performed at 110–140 °C for times ranging between 6 and 10 h and alkalization do not completely eliminate spore-forming bacteria and, therefore, reach commercial sterility [36,53]. A reduction of 4 log CFU/g of the initially inoculated spores has been observed at most [10,36,53]. Additional reduction can be obtained by the combination of roasting with steam treatment capable of eliminating spore-forming bacteria [54,55], which can reduce the populations of vegetative bacteria by more than 6 log units [54].

Therefore, as previously reported by the aforementioned authors, it may be concluded that the absence of non-spore-forming microorganisms in the cocoa powder analyzed in this study is due to the production process and, in particular, to the husking and roasting phases, which are both critical control points (CCPs) since, after the latter, cocoa is considered completely safe at the microbiological level due to the temperature conditions and duration of the process [4]. Finally, when applying the cocoa industrial powder guidelines [56], which include specifications for total aerobic mesophiles (<5000 CFU/g), molds (<50 CFU/g), yeasts (<10 CFU/g), *Enterobacteriaceae* (absent in 1 g) and *Salmonella* (absent in 25 g), it was concluded that all investigated cocoa powders are largely acceptable.

### 3.2. Chocolate Contamination

The chocolate samples of the different brands showed heterogeneous contamination (Table 7). In particular, the thermophilic spore-forming microorganisms were found at levels below the quantification limit of the method (<1 CFU/g), while the mesophilic spore-forming microorganisms were determined at concentrations higher than the minimum quantification limit in all brands analyzed. However, the concentration of these mesophilic spores was low and always lower than 1.5 log CFU/g and between 15 and 30 CFU/g. The mold concentration divided the brands into two different groups (*p* < 0.05). The results highlight a group including the samples of brands 1, 2 and 3 with an average fungal concentration between 1.2 and 1.3 log CFU/g and a second group including the samples of brands 4 and 5 with an average fungal concentration of approximately 1.5 log CFU/g. In cocoa powder, *Salmonella* spp., *E. coli* and *Listeria monocytogenes* were never isolated. In particular, *Salmonella* was absent in 25 g of each chocolate sample. This eliminates any doubt concerning the unhealthiness of these products; in the literature, it is reported that even low concentration levels can lead to cases of infection in the consumer [57]. In fact, *Salmonella* spp. represents a real problem in chocolate because, over the last forty years, numerous cases of salmonellosis have been described, despite the evolution of production systems. For this reason chocolate is subject to official sampling (EU Reg. 2017/625, http://data.europa.eu/eli/reg/2017/625/oj, consultation 9 August 2022) included in Multi Annual National Control Plans, investigating food safety criteria (at retail level) and process hygiene criteria (official controls at food establishments) according to the provisions of EC Reg. 2073/2005 (http://data.europa.eu/eli/reg/2005/2073/oj, Consultation 9 August 2022). Considering the absence of *Listeria monocytogenes* and *Salmonella* spp. in all the chocolate samples, the microbial criteria of EC. Reg. 2073/2005 are respected.

In addition, *Salmonella* spp. cannot grow in chocolate due to the low Aw (0.3–0.5) and high-fat concentration (>20%); however, these parameters allow its survival for long periods [58] and its ability to withstand heat treatments [59]. Furthermore, fat acts as a protector against the low pH of the gastric juice and, therefore, even low concentrations of this microorganism can stress the possibility of remaining vital during digestion as evidenced in the salmonellosis caused by *Salmonella* Nima infection, whose concentration was 0.043 MPN/g of chocolate [60]. *Enterobacteriaceae* also were never detected in the analyzed chocolate samples (<1 CFU/g). In contrast, the presence of both microorganisms has already been reported by several authors [61]. In their study, fecal markers, such as *E. coli,* were also isolated at levels of 0.6–1.1 log MPN/g in 7% of the analyzed samples. Different concentrations of strains belonging to *Enterobacteriaceae* have often been enumerated and identified in chocolate from different international companies [20,61,62,63]. The presence of *Enterobacteriaceae*, including *Salmonella* spp., in the finished product results from the hands of the workers and the equipment used and therefore, this fact is considered an indication of poor hygiene in the chocolate industry [62]. The origin is rarely attributable to the raw materials because the roasting of the cocoa beans and the chocolate conching phase, which is performed at 50–80 °C, can contribute to a reduction or elimination of *Enterobacteriaceae* [5]. In addition, *Salmonella* and *E. coli* are rarely found in cocoa or cocoa plantations because both are of human fecal origin and, consequently, can be introduced at some point during the process through the hands of the workers who manipulate the fruits and turn the beans during drying [4,55]. Therefore, the presence of *Enterobacteriaceae*, *Salmonella,* or other vegetative microorganisms can only occur during the post-conching phases, i.e., cooling, molding, demolding and packaging of the chocolate, and originates from the added ingredients, contaminated processing machinery, or the surroundings [4]. Nascimiento et al. [62] isolated *Enterobacteriaceae* and thermotolerant coliforms from equipment and utensils and demonstrated that the manufacturing environment, including food handlers, must be considered the most likely contamination source of the finished product. Therefore, the ICMSF [54] recommended research investigating *Enterobacteriaceae* in the equipment, product residues in contact with surfaces, production environment and finished products and, in particular, the application of good hygiene practices (GHP), which are designed to prevent post-process contamination and, thus, are necessary for ensuring a clean and safe product. However, it is of great importance to research *Salmonella* spp. because the lack or the low concentration of *Enterobacteriaceae* and total coliforms cannot exclude its presence [64]. In addition, *Salmonella* can survive conch and other treatments because its thermal resistance increases in low-moisture high-sugar products [65,66].

The temperature used during the conching phase of chocolate also eliminates yeasts [4], which were not isolated in this study.

The main microorganisms observed in the investigated chocolates belong to species that were already isolated in cocoa powder. Colonies of *B. subtilis* and *B. licheniformis* were isolated at ratios of 60.4% and 29.4%, respectively. Such microorganisms can be derived from cocoa beans and are only subjected to a minimal reduction during roasting as already reported by several authors [4,10]. *Bacillus flexus* (*Priesta flexa*) and *Jeotgalibacillus marinus* have also been isolated, albeit to a lesser extent. Both can be identified as accidental contaminants as they have never been described in any phase of chocolate production. The isolated molds were identified through traditional and molecular methods (Table 2). As can be observed, penicilli predominate, particularly *P. lanosocoeruleum*, which represents 53.3% of the isolates, followed by *Cladosporium cladosporioides* (36.7%), *P. brevicompactum* and *P. chrysogenum*, which were isolated in 5% of the cases. It is believed that, considering the isolated species, the contamination may have an environmental origin. In fact, the filamentous fungal species identified during the fermentation, dehydration and storage of beans are completely different from those isolated in this study. According to Copetti et al. [50], the contamination of broad beans consists of most species belonging to the genera *Aspergillus*, *Eurotium*, *Mucor*, *Absidia and Rhizopus*, which are species of environmental origin and types of plants that are generally able to develop at low Aw with low nutritional requirements. In particular, as occurs in cereals, nuts and spices [25,67], there is an increase in these microorganisms during the storage of fermented beans when the xerophilic or xero-tolerant species can take over [4,50]. In contrast, inadequate storage causes beans to absorb up to 8% of the humidity from the surrounding environment, enabling the development of *Aspergillus* section *Flavi* and *Nigri*, which are responsible for the production of ochratoxins [68].

*Penicillium* and *Cladosporium* species seem to be present in fermented beans in a discontinuous way in different concentrations compared to the typical cases mentioned above. However, these indigenous species are completely inactivated by roasting and conching during the chocolate production process [50], further confirming that the isolated molds derived from the post-conching chocolate manufacturing environment are chocolate-based products, e.g., pralines and coated nuts, which become moldy due to their high Aw (>0, 70). In this type of product, species belonging to the genera *Aspergillus* and especially *Penicillium* spp. have often been isolated and their origin is strictly related to the raw materials used (syrups, sugar, vanilla, etc.), the production environment and, partially, the chocolate used. In fact, in these substrates, the following species have often been isolated at the level of 5 log CFU/g: *Aspergillus oryzae*, *Aspergillus niger*, *P. corylophilum*, *Penicillium chrysogenum*, *P. brevicompactum*, *A. terreus*, *A. tubigensis*, *Eurotium asmtelodami* and *E. repens* and xerophilic yeasts, such as *Zygosaccharomyces* [17,20].

In previous studies, some xerophilic fungi have also been isolated from manufactured chocolate products [13]. *Bettsia alvei* (teleomorph of *Chrysosporium farinicola*) and *Chrysosporium xerophilum* were isolated from spoilt hazelnut chocolate, *Neosortorya glabra* was isolated from spoilt chocolate confectionery and *Chrysosporium farinicola* was isolated from chocolate [13,16]. Kinderlerer [13] reported that, although the low Aw values of industrial and commercial samples are not conducive to microbial growth, xerophilic fungi, such as *Chrysosporium* spp. and *C. inops,* represent potential spoilage organisms of chocolate and chocolate products [13,69]. However, the spoilage was usually due to a fat bloom originating from recrystallization of the cocoa butter on the outer and inner chocolate surface and inadequate storage problems arising from the formation of an environment with an increased availability of water at the interface of the packaging and chocolate when it was stored in environments with high relative humidity [13,69].

### 3.3. Mouldy Chocolates

The hazard that some filamentous fungi can grow on chocolate bars has been confirmed by the collection of five samples of milk chocolate filled with milk cream in which fungal development was evident (Figure 1). The bars belonged to one of the investigated brands collected in a small shop and the strains responsible for the spoilage were *P. lanosocoeruleum* species. The spoiled chocolates had an Aw of 0.50, a value that did not allow the development of the same species in the synthetic culture medium used (Table 9). Indeed, as observed, only *P. lanosocoeruleum* grows in substrates with an Aw of 0.80 but does not grow at a lower Aw as demonstrated also by other authors [10]. Filamentous fungi and yeast development were also observed on chocolate pralines [17], which have an Aw > 0.80. The pH did not influence the development of the molds. Other fungi strains grew at Aw 0.90 and *Bacillus* spp. never grew at each tested Aw.

It is assumed that *P. lanosocoeruleum* may have developed in chocolate for many reasons. The first is due to a superficial presence of moisture absorbed during demolding, packaging, or storage. Notably, chocolate is a hygroscopic product that can acquire humidity and, therefore, become a substrate for xerophilic or xero-tolerant molds and yeasts [50,70].

The second reason is the fat bloom phenomenon accompanied by changes in the crystallization of cocoa butter on the chocolate surface [13], which introduces moisture to the surface. Most likely, the molds that contaminate the surface have taken advantage of this situation, developing and producing an evident patina.

### 3.4. Growth of P. lanosocoeruleum in Agars and Chocolate Bars

The potential spoilage capacity of *P. lanosocoeruleum* to grow on chocolate bars was tested in agars and directly in chocolate bar. The growth of the molds in all samples incubated at 20 and 30 °C was considered positive at the time (days) when hyphae or spores were visible. Table 3 shows the growth of the strain in different chocolate model systems stored in air and in vacuum packaging at 20 and 30 °C. As shown at both temperatures, no growth was observed in vacuum packaging and in air on EBC at Aw levels less than 0.712, EFMC less than 0.732 and MCFC less than 0.748. The atmosphere (air/under vacuum) and type of chocolate influenced the growth more regardless of the temperature. Indeed, the times spent growing up depended on the types of chocolates and ranged between 92 and 52 days at 20 °C and 90 and 50 days at 30 °C. As expected, *P. lanosocoeruleum* grew faster in MCFC and EFMC than in EBC at both temperatures probably due to the different Aw levels and the recipes of the three types of chocolate bars. Indeed, initially, the Aw of EBC was approximately 0.510 ± 0.006, the Aw of EFMC was 0.530 ± 0.02 and the Aw of MCFC was 0.550 ± 0.003. Previous works confirmed the influence of air, the Aw and the recipe on the growth of xerophilic fungal species on chocolate [13,17]. In particular, it is well known that osmophilic yeasts, xerophilic molds and bacteria that tolerate low Aw cause problems in milk chocolates and chocolate fillings and produce spoilage represented by off-flavor, slime formation and gas production that leads to cracking of the chocolates or visible growth on the surface of the liquid fraction [17,18,71]. As expected, *Penicillium lanosocoeruleum,* a strictly aerobic mold, did not grow in the atmosphere without oxygen. In contrast, previous works have demonstrated that some food-borne mold species, such as *Chryptosporium* spp., can grow in reduced oxygen concentrations presumably due to their fermentative metabolism [13]. In particular, Kinderlerer [13] demonstrated that *C. inops* produced ethanol and carbon dioxide, growing inside products, such as chocolate-covered confectionery products. However, the author admits that the growth depended on the strain, which was able to live in a low Aw and the presence of the recrystallization of the cocoa butter in the spoiled chocolate. Finally, the growth of *P. lanosocoeruleum* on the three types of chocolates incubated at 20 and 30 °C in air and at different ERHs (equilibrium relative humidity) was evaluated (Table 4). Additionally, in this experiment, the chocolate bars included extra bitter chocolate (EBC, Aw 0.510 ± 0.006), extra fine milk chocolate (EFMC, Aw 0.530 ± 0.02) and milk chocolate filled with milk cream with added vanilla flavor (MCFC, Aw 0.550 ± 0.003). As shown in Table 4, the fungal growth started to be visible at ERH 90%. The times of growth depended on the recipe, temperature and ERH and ranged between 26 and 74 days. The strain grew faster in MCFC and EFMC than in EBC. The growth depended on the amount of water that condensed on the chocolate, which was derived from the different solutions. Indeed, the water vapor originating in the closed boxes by the solutions was adsorbed by the chocolates, being hygroscopes and allowed *P. lanosocoeruleum* growth. The results confirmed previous works that demonstrated the influence of water availability on mold growth in chocolate [50]. Indeed, chocolate spoilage occurs in episodes during which inadequate storage problems arise from the formation of an environment with an increased availability of water at the interface of the packaging and chocolate when it is stored in environments with high relative humidity [13,50,69]. Therefore, the temperature of the chocolate packing room is usually maintained between 18 and 20 °C, with a relative humidity of less than 50% to prevent condensation of the final products as they leave the cooling tunnel [50,72]. In addition, fluctuations in temperature must be avoided during storage as these fluctuations can accelerate the formation of an opaque appearance, fat bloom and, consequently, mold growth.

## 4. Conclusions

The microbial population observed in cocoa powder and chocolate bars of three batches of five different brands sold in Italian markets is varied and represented by mesophilic and thermophilic spore-forming bacteria and molds. The cocoa powder did not show the presence of the molds that were isolated from the bars, while the latter did not contain thermophilic spore-forming microorganisms. The origin of these microorganisms is considered closely linked to the product. In fact, in cocoa and chocolate bars, the isolated spore-forming microorganisms probably derive from the raw material as they can survive the extraction and manufacturing processes of the cocoa supply chain. In fact, the same species have been extensively isolated from raw cocoa beans by other authors and are heat resistant; they survived the roasting of the beans and the conching of the chocolate. Although these treatments are performed at a high temperature (roasting 110–140 °C; conching 50–80 °C), they do not result in commercial sterility. Conversely, the origin of molds in chocolate is related to the production environment and occurs in the final post-conching phases (cooling and packaging). In fact, the times and temperatures of conching allow the elimination of any non-spore-forming microorganisms, including molds and yeasts. However, both types of contaminants have a limited concentration in the final product, are always less than 2 log CFU/g and do not develop in both products due to their low Aw (<0.6), consequently they do not influence either the spoilage or the safety. A case of mold development in chocolate bars was observed in this study. This growth can be explained by a cocoa butter bloom accompanied by the presence of humidity derived from the bloom or acquired during packaging. The isolated species, *Penicillium lanosocoeruleum*, responsible for the spoilage is xerophilic. Its growth was confirmed in vitro and in vivo and depended on the chocolate recipe, temperature, flat bloom and Aw. In each case, growth occurred in air, at an Aw level over 0.80 and EHR over 90% or with water condensed on the chocolate.

Finally, to reduce or eliminate microorganisms contaminations from cacao or chocolate it is necessary to apply the HACCP/Food Safety Management System (Hazard Analysis Critical Control Point) either online or offline. In particular it must be checked: online, the time/temperature of production and, offline, cleaning and disinfection, pests and unwanted animals, water potability, the maintenance of facilities and processing plants, the ambient and storage temperatures, staff training, health and hygiene, the management of by-products, waste and effluents and emissions into the atmosphere, the selection and verification of suppliers, the procedures for assigning lots and traceability, the recall and withdrawal from the market of non-compliant products, the definition of the shelf life of products and the management of labels and identification marks.

## Figures and Tables

**Figure 1 foods-11-02753-f001:**
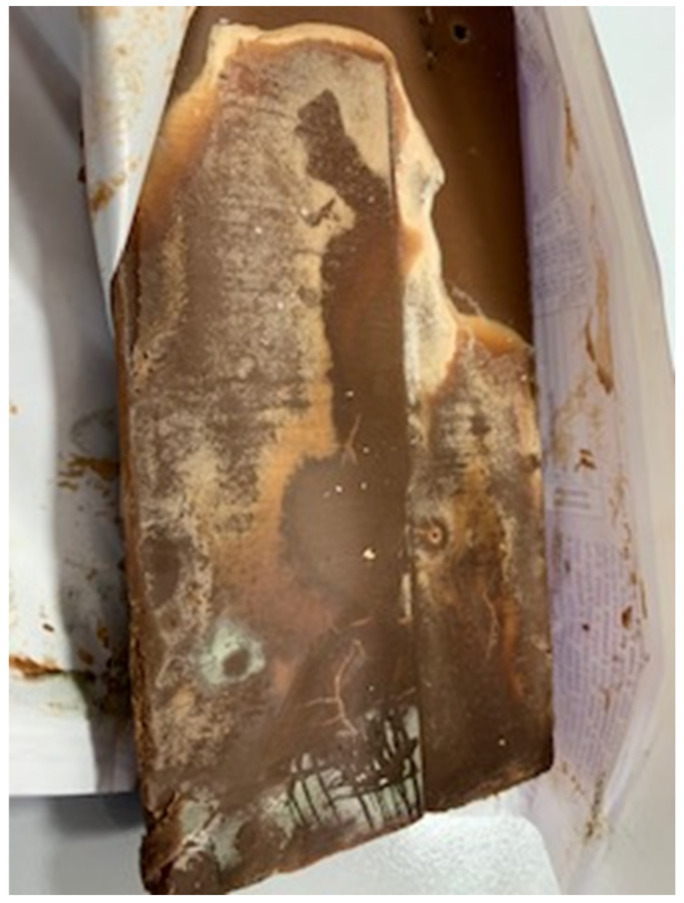
Moldy chocolate.

**Table 1 foods-11-02753-t001:** Bacterial species isolated from chocolate.

Microorganisms	Accession Number	Brand					
		1	2	3	4	5	%
*Bacillus subtilis*	NR_112116.2	66	66	53	70	47	60.4
*Bacillus licheniformis*	MK515041.1	18	34	32	30	33	29.4
*Bacillus flexus* *	MT255044.1	16		12		20	9.6
*Jeotgalibacillus marinus*	MK439548.1			3			0.6

* *Priestia flexa;* Total Number of the isolated strains: 500; % isolated.

**Table 2 foods-11-02753-t002:** Fungal species isolated from chocolate.

Microorganisms	Accession Number	Brand					
		1	2	3	4	5	%
*Penicillium brevicompactum*	MH876753.1					6	5.0
*Penicillium lanosocoeruleum*	MH873469.1	15	10	9	17	13	53.3
*Penicillium chrysogenum*	MH877425.1		3			3	5.0
*Cladosporium cladosporioides*	JF499855.1	9	11	15	7	2	36.7

Total Number of isolated strains:120; % isolated.

**Table 3 foods-11-02753-t003:** Growth of *Penicillium lanosocoeruleum* in chocolate bars at different Aw levels.

Chocolate Bars	Aw	Time to Detect the Hyphas/Spores (Days)	
		Air Packaging	Vacuum Packaging
Extra bitter	0.510 ± 0.006	No growth	No growth
	0.602 ± 0.003	No growth	No growth
	0.712 ± 0.005	No growth	No growth
	0.835 ± 0.002	92 ± 3 a/90 ± 5 a	No growth
Extra Fine Milk	0.530 ± 0.002	No growth	No growth
	0.628 ± 0.005	No growth	No growth
	0.732 ± 0.003	No growth	No growth
	0.840 ± 0.002	85 ± 3 a/83 ± 1 a	No growth
Milk filled Cream	0.550 ± 0.002	No growth	No growth
	0.655 ± 0.003	No growth	No growth
	0.748 ± 0.002	No growth	No growth
	0.862 ± 0.004	52 ± 7 a/50 ± 5 a	No growth

T/T: 20/30 °C. Data were represented as the mean ± standard deviation; Different letters indicate significant differences between the two temperatures within the same chocolate type (*p* < 0.05).

**Table 4 foods-11-02753-t004:** *Penicillium lanosoceruleum* growth on chocolate bars stored in different ERH% at 20 and 30 °C.

ERH %	NaCl %	Time to Detect the Hyphas/Spores (Days)		
		Extra Bitter	Extra Fine Milk	Milk Filled Cream
75	35.70	No growth	No growth	No growth
81	22.50	No growth	No growth	No growth
86	18.18	No growth	No growth	No growth
90	14.18	74 ± 2 a/72 ± 3 a	65 ± 3 a/63 ± 2 a	62 ± 1 a/61± 2 a
94	9.38	65 ± 3 a/60 ± 5 a	55 ± 3 a/53 ± 5 a	53 ± 3 a/52± 6 a
99	1.74	30 ± 4 a/28 ± 4 a	28/± 4 a/26 ± 5 a	28/± 4 a/26± 3 a

T/T: 20/30 °C. Data were represented as the mean ± standard deviation; Different letters indicate significant differences between the two temperatures within the same chocolate type (*p* < 0.05).

**Table 5 foods-11-02753-t005:** Chemical and physical characteristics of cocoa powder and chocolate bars.

Brand	Cocoa Powder		Chocolate	
	pH	Aw	pH	Aw
1	6.9 ± 0.2 a	0.406 ± 0.003 a	6.8 ± 0.3 a	0.527 ± 0.005 a
2	5.7 ± 0.1 b	0.389 ± 0.005 b	5.6 ± 0.2 b	0.512 ± 0.010 b
3	6.3 ± 0.5 ab	0.409 ± 0.004 a	6.1 ± 0.2 c	0.530 ± 0.006 a
4	7.1 ± 0.2 a	0.390 ± 0.003 b	7.0 ± 0.1 a	0.520 ± 0.001 c
5	5.3 ± 0.1 d	0.412 ± 0.007 c	5.5 ± 0.2 b	0.522 ± 0.003 c

Mean ± standard deviation; Means in the same column with the same letters are not significantly different (*p* < 0.05).

**Table 6 foods-11-02753-t006:** Microbial population of cacao powder of 5 different brands sold in the Italian market.

Temperature	30 °C	55 °C	30 °C	55 °C	25 °C
Brand	CBT	CBT	CBT Spores	CBT Spores	Fungi
1	1.1 ± 0.4 aw	1.0 ± 0.1 aw	1.3 ± 0.3 aw	1.1 ± 0.2 aw	<1 ax
2	1.4 ± 0.1 aw	1.2 ± 0.2 aw	1.5 ± 0.3 aw	1.5 ± 0.1 bw	<1 ay
3	1.2 ± 0.4 aw	1.2 ± 0.3 aw	1.3 ± 0.2 aw	1.2 ± 0.2 abw	<1 ax
4	1.2 ± 0.3 aw	<1 bx	1.1 ± 0.1 aw	<1 bx	<1 ax
5	1.3 ± 0.1 aw	<1 bx	1.2 ± 0.3 aw	<1 bx	<1 ax

Data mean ± standard deviation-log CFU/g; <1 CFU/g; CBT: Total microbial count; Fungi: yeasts and molds; Means with the same letters in the same column (a,b and in the same row (w,x,y, considering each parameter, are not significantly different (*p* < 0.05).

**Table 7 foods-11-02753-t007:** Microbial population in chocolate of 5 different brands sold in the Italian market.

Temperature	30 °C	55 °C	30 °C	55 °C	25 °C
Brand	CBT	CBT	CBT Spores	CBT Spores	Molds
1	1.1 ± 0.1 aw	<1 ax	1.2 ± 0.1 aw	<1 ax	1.3 ± 0.2 aw
2	1.3 ± 0.1 aw	<1 ax	1.1 ± 0.3 aw	<1 ax	1.2 ± 0.2 aw
3	1.1 ± 0.2 aw	<1 ax	1.2 ± 0.2 aw	<1 ax	1.3 ± 0.1 aw
4	1.3 ± 0.1 aw	<1 ax	1.1 ± 0.3 aw	<1 ax	1.5 ± 0.2 bw
5	1.3 ± 0.1 aw	<1 ax	1.3 ± 0.2 aw	<1 ax	1.5 ± 0.1 bw

Data mean ± standard deviation-log CFU/g; < 5 CFU/g; CBT: Total microbial count; Yeasts; < 1 CFU/g. Means with the same letters in the same column (a,b and in the same row (w,x, considering each parameter, are not significantly different (*p* < 0.05).

**Table 8 foods-11-02753-t008:** Bacterial species isolated from cocoa powder.

Microorganisms	Accession Number	Brand					
		1	2	3	4	5	%
*Bacillus subtilis*	NR_112116.2	55	63	58	65	50	58.2
*Bacillus licheniformis*	MK 515041.1		8		3	15	5.2
*Bacillus amyloliquefaciens*	NR_041455.1	26	15	28	8	10	17.4
*Geobacillus stearothermophilus*	NR_115284.2	19	14	14	24	25	19.2

Total number of the isolated strains 500; % isolated.

**Table 9 foods-11-02753-t009:** Growth of different microbial species isolated from cocoa powder and chocolate in Malt Agar and * Plate Count Agar with different Aw and pH.

Species	Aw			
	0.90	0.85	0.80	0.75–0.50
*Penicillium brevicompactum*	2.8 ± 0.5	n.g.	n.g.	n.g.
*Penicillium lanosocoeruleum*	5.5 ± 1.0	2.5 ± 0.4	1.1 ± 0.5	n.g.
*Penicillium chrysogenum*	2.7 ± 0.5	n.g.	n.g.	n.g.
*Cladosporium cladosporioides*	2.4 ± 0.5	n.g.	n.g.	n.g.
*Bacillus* spp. *	n.g.	n.g.	n.g.	n.g.

Data mean ± standard deviation; Growth in cm/50 days; Malt Agar with 50% of glucose and glycerol; n.g. no growth; * all the isolated species in PCA 50% glucose–growth at 30 and 55 °C.

## Data Availability

Data are contained within the article.

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
