# Peer review of "Microbial Characterization of Retail Cocoa Powders and Chocolate Bars of Five Brands Sold in Italian Supermarkets"

_foods, 2022, doi:10.3390/foods11182753_

Round 1
Reviewer 1 Report
The study is quite impressive and scientifically elaborated. I advise some major corrections:
- In the Sample collection and microbiological analysis section, How to differentiate between yeasts and bacteria on Petri dishes containing malt extract agar? is it necessary to use a broad-spectrum antibacterial (example chloramphenicol) to kill all bacteria and leave only yeasts and moulds.
- In the material and methods section, please add the description of the compounds used and all protocols to make the essays more visible for lecturers.
- I suggest for authors to verify the parameters and conditions of the analysis and the machine information (name, manufacturers ...).
- Ensure the uniformity in the units (e.g. mg/mL, µL) throughout the MS.
- Correct the English mistakes. Spelling should be revised thoroughly, many spelling mistakes detected.
- Check the references in accordance with the journal style.
Author Response
Response to Reviewer 1 Comments
Manuscript ID - Foods-1900934
Microbial Characterization of Retail Cocoa Powders and Chocolate Bars of Five Brands Sold in Italian Supermarkets
Journal: Foods
Answer to the referee.
The authors would like to thank the reviewers for their careful reading of the manuscript and the resulting constructive comments and suggestions. Basically, we agree with all of the points raised by the reviewers, and wherever possible the manuscript has been modified as recommended. All reviewer comments are in black plain font, whereas our response is described in red plain font.
We have made the changes and corrections on the basis of the reviewer’s suggestions. We evaluated the comments and prepared a point-by-point response to each one of them.
Comments from the editors and reviewers:
The manuscript has been modified including most of the suggestions. Then, it can be now accepted for publication.
Reviewers' comments:
The study is quite impressive and scientifically elaborated. I advise some major corrections:
- In the Sample collection and microbiological analysis section, How to differentiate between yeasts and bacteria on Petri dishes containing malt extract agar? is it necessary to use a broad-spectrum antibacterial (example chloramphenicol) to kill all bacteria and leave only yeasts and moulds.
Thanks – Answer – Line 141 - I forgot to cite the addition of Chloramphenicol (0.100 g/L) Sigma–Aldrich, Milan, Italy.
- In the material and methods section, please add the description of the compounds used and all protocols to make the essays more visible for lecturers.
- I suggest for authors to verify the parameters and conditions of the analysis and the machine information (name, manufacturers ...).
Thanks – Answer – I add the follows parts for both the suggestions
Lines 155-164 - Listeria monocytogenes – (Briefly – 25 g product were added to 225 ml of Fraser broth (Oxoid, Italy) incubated at 30 °C for 24 h, then an aliquot of this broths was streaked on Chromocult Listeria Agar according to Ottaviani/Agosti agar (Biolife, Milan, Italy) incubated at 37 °C for 24 h. On this agar L. monocytogenes produce typical blue-green colonies surrounded by an opaque halo);
Salmonella (briefly: 25 g product were added to 225 ml of Buffered Peptone Water (BPW, Oxoid, Italy) incubated 18 h at 37 °C, then 1 ml of BPW in 9 ml of Rappaport Vassiliadis broth (RVB, Oxoid, Milan, Italy) incubated at 42 °C for 18-24 h. An aliquot of RVB was streaked on Xylose Lysine Tergitol 4 agar (Oxoid, Milan, Italy) incubated at 37 °C for 24 h. On this agar the black or center black colonies were presumptive Salmonella.
Lines 166-211
From the PCA plates of the cocoa powder and chocolate samples, 100 colonies, selected independently from their morphology, color, and size were isolated from each brand considered. After purification, the colonies were identified according to the PCR-DGGE (PCR-denaturing gradient gel electrophoresis) method as described by Iacumin et al. [22]. Briefly, amplicons to be subjected to DGGE analysis were obtained using primers P1 (5’-GCGGCGTGCCTAATACATGC-3’) and P4 (5’ ATCTACGCATTTCACCGCTAC-3’) spanning the V3 region of the 16S rDNA [23]. A GC‐clamp (5′‐CGC CCG CCG CGC CCC GCG CCC GTC CCG CCG CCC CCG CCC G‐3) was attached to the 5’ end of primer P1. Amplifications were carried out in a final volume of 25 μl, containing 1 μl (100 ng total) template DNA, 10 mM Tris HCl (pH 8.3), 50 mM KCl, 1.5 mM MgCl2, 0.2 mM deoxynucleoside triphosphates (dNTPs), 1.25 U Taq polymerase (Invitrogen, Milan, Italy), and 0.2 μM each primer, using the C1000 Touch Thermal Cycler (Bio-Rad, Milan, Italy). The amplification cycle included an initial denaturation step at 95 °C for 5 min, followed by 35 series composed by denaturation, performed at 95 °C for 1 min, annealing at 45 °C for 1 min, and extension performed at 72 °C for 1 min. Finally, an extension cycle, performed at 72 °C for 7 min, was added. Electrophoresis was performed in a 0.8‐mm‐thick polyacrylamide gel (8% (wt/vol) acrylamide bisacrylamide (37.5:1)), with a denaturing gradient from 30% to 50% (100% corresponded to 7 M urea and 40% (wt/vol) formamide) increasing in the direction of the electrophoretic run using the Dcode universal mutation detection system (Bio-Rad, Hercules, California). Gels were subjected to a constant voltage of 130 V for 3 h and 30 min at 60 °C. After the electrophoresis, gels were stained for 30 min in 1.25X Tris‐acetate‐EDTA containing 1X SYBR Green (final concentration; Molecular Probes, Milan, Italy). Pictures of the gels were visualized under UV light using the Syngene G: Box Chemi‐XX9 (Syngene, Cambridge, United Kindom) and digitally captured by using the software GeneSys version 1.5.7.0 (Syngene, Cambridge, United Kindom). Strains with the same DGGE profile were grouped, and 2 representatives per group were amplified using the primers P1 and P4 targeting 700 bp of the V1-V3 region of the 16S rRNA gene (rDNA). Amplifications were carried out as reported above. After purification, amplicons were sent to a commercial facility for sequencing (Eurofins AG, Ebersberg, Germany). The sequences were aligned in GenBank using the Blast program version 2.13.0 [24] to determine the closest known relatives of the partial 16S rDNA sequence obtained.
From the MEAG plates of each chocolate brand, 24 mold colonies were isolated and transplanted into the following three different medium cultures: Czapek Dox Agar (Oxoid, Milan, Italy), MEA and Malt Salt agar (5% malt extract; 5% NaCl; pH 6.2; Oxoid, Milan, Italy). The isolated molds were identified according to the traditional methods proposed by Samson et al. [25]: examination of macroscopic (colonial) and microscopic characteristics of the molds grown on the three agars. In particular, morphology, shape, size, and color of the colonies and spores, production of pigment, as well as type, development and texture of the mycelium were observed. The presuntive identification was then confirmed by sequencing a fragment of ~600 bp of the D1-D2 region of the large-subunit rRNA gene according to the methods reported by Iacumin et al. [22]. Primers NL1 (5’-GCC ATA TCA ATA AGC GGA GGA AAA G-3’) and NL4 (5’-GGT CCG TGT TTC AAG ACG G-3’) were used. Reaction mixture and amplification protocol were the same as those described above. Sequence comparisons were performed in GenBank using the Blast program version 2.13.0.
- Ensure the uniformity in the units (e.g. mg/mL, µL) throughout the MS.
Thanks – Answer I correct all.
- Correct the English mistakes. Spelling should be revised thoroughly, many spelling mistakes detected.
Thanks – Answer – I made it
- Check the references in accordance with the journal style.
Thanks – Answer I add the doi to complete the references

Reviewer 2 Report
Lucilla Iacumin et al. submitted to Foods an article focusing on the microbial characterization of five different brands of chocolate bars and cocoa powders, sold in italian retails.
The manuscript is useful for experts in this field, both to better evaluate the preventive actions to be taken in food establishments, and to improve the official control phases related to food safety. The paper is written according to a scientific logic, is supported by an adequate rich bibliography and has been drawn up meticulously, following the Journal rules. Reading in English is fluid and understandable.
Here are my suggestions to improve this paper:
- in addition to “microbial”, please evaluate if "physico-chemical" is to be added in the title;
- in the discussion section it could be useful to mention that “chocolate” as food matrix (meaning as “ready to eat”) is subject to official sampling (EU Reg. 2017/625) included in Multi Annual National Control Plans, investigating food safety criteria (at retail level) and process hygiene criteria (official controls at food establishments) according to the provisions of EC Reg. 2073/2005;
- is it possible to hypothesize that also the storage temperatures of the raw material (cocoa beans) in the long phases of sea transport can interfere and create favorable environments to microbial replication / fungal reproduction (e.g. hot / humid or cold / humid environments or thermal stress) inside the containers or cargo hold? If so, it would be interesting to mention it in discussions, supported by citations.
- to make the manuscript more attractive for Public Health experts, it would be appropriate to specify the useful preventive actions to be taken in Food Business Operator’s own checks (HACCP / Food Safety Management System) to prevent contamination of environmental origin in the process phases at food businesses (storage, cleaning, sanitizing of work lines and conveyor belts, maintenance of filters of ventilation and air conditioning systems, pest control, etc.).
Thank you for your efforts in perfecting this interesting article.
Author Response
Response to Reviewer 2 Comments
Manuscript ID - Foods-1900934
Microbial Characterization of Retail Cocoa Powders and Chocolate Bars of Five Brands Sold in Italian Supermarkets
Journal: Foods
Answer to the referee.
The authors would like to thank the reviewers for their careful reading of the manuscript and the resulting constructive comments and suggestions. Basically, we agree with all of the points raised by the reviewers, and wherever possible the manuscript has been modified as recommended. All reviewer comments are in black plain font, whereas our response is described in red plain font.
We have made the changes and corrections on the basis of the reviewer’s suggestions. We evaluated the comments and prepared a point-by-point response to each one of them.
Comments from the editors and reviewers:
The manuscript has been modified including most of the suggestions. Then, it can be now accepted for publication.
Reviewers' comments:
The manuscript is useful for experts in this field, both to better evaluate the preventive actions to be taken in food establishments, and to improve the official control phases related to food safety. The paper is written according to a scientific logic, is supported by an adequate rich bibliography and has been drawn up meticulously, following the Journal rules. Reading in English is fluid and understandable.
Here are my suggestions to improve this paper:
- in addition to “microbial”, please evaluate if "physico-chemical" is to be added in the title;
Thanks – Answer – We think it was unnecessary.
- in the discussion section it could be useful to mention that “chocolate” as food matrix (meaning as “ready to eat”) is subject to official sampling (EU Reg. 2017/625) included in Multi Annual National Control Plans, investigating food safety criteria (at retail level) and process hygiene criteria (official controls at food establishments) according to the provisions of EC Reg. 2073/2005;
Thanks – Answer – I add in the discussion For this reason chocolate is subject to official sampling (EU Reg. 2017/625, http://data.europa.eu/eli/reg/2017/625/oj, consultation 2022, August) included in Multi Annual National Control Plans, investigating food safety criteria (at retail level) and process hygiene criteria (official controls at food establishments) according to the provisions of EC Reg. 2073/2005 (http://data.europa.eu/eli/reg/2005/2073/oj, Consultation 2022, August). Considering the absence of Listeria monocytogenes and Salmonella spp. in all the chocolate samples, the microbial criteria of EC. Reg. 2073/2005 are respected.
- is it possible to hypothesize that also the storage temperatures of the raw material (cocoa beans) in the long phases of sea transport can interfere and create favorable environments to microbial replication / fungal reproduction (e.g. hot / humid or cold / humid environments or thermal stress) inside the containers or cargo hold? If so, it would be interesting to mention it in discussions, supported by citations.
Thanks – Answer – Usually sea transport of the raw material does not produce microorganisms growth, because the beans are dried (< 8% moisture) and are kept in containers subjected to different treatments such as fumigation such as with methyl bromide (Ferrari, 2014; Copetti et al., 2011).
- to make the manuscript more attractive for Public Health experts, it would be appropriate to specify the useful preventive actions to be taken in Food Business Operator’s own checks (HACCP / Food Safety Management System) to prevent contamination of environmental origin in the process phases at food businesses (storage, cleaning, sanitizing of work lines and conveyor belts, maintenance of filters of ventilation and air conditioning systems, pest control, etc.).
Thanks – Answer – I add Finally to reduce or eliminate microorganisms contaminations from cacao or chocolate it is necessary to apply the HACCP/Food Safety Management System (Hazard Analysis Critical Control Point) either on line either of line. In particular it must be checked: on line, the time/temperature of production and off-line, the cleaning and disinfection, the pest and unwanted animals, the water potability, the maintenance of facilities and processing plants, the ambient and storage temperatures, the staff training, health and hygiene, the management of by-products, of waste and effluents and emissions into the atmosphere, the selection and verification of suppliers, the procedures for assigning lots and traceability, the recall and withdrawal from the market of non-compliant products, the definition of the shelf life of products and the management of labels and identification marks.

Round 2
Reviewer 1 Report
Many thanks for your answers.